# Machine Learning Sorting Method of Bauxite Based on SE-Enhanced Network

**Pengfei Zhao [1], Zhengjie Luo [1], Jiansu Li [1], Yujun Liu [2] and Baocheng Zhang [3],***

[1] Department of Mechanical and Electronic Engineering, School of Mechanical Engineering, North University of China, No. 3 Xueyuan Road, Taiyuan 030051, China; wings215@nuc.edu.cn (P.Z.); s202106129@st.nuc.edu.cn (Z.L.); jslihongcha@126.com (J.L.)

[2] China Unicom Taiyuan Branch, Taiyuan 030001, China; lixinxjx@163.com

[3] Department of Mechatronics Engineering, School of Engineering, Ocean University of China, Qingdao 266100, China

* Correspondence: zbc2014088@ouc.edu.cn

**Abstract:** A fast and accurate bauxite recognition method combining an attention module and a clustering algorithm is proposed in this paper. By introducing the K-means clustering algorithm into the YOLOv4 network and embedding the SE attention module, we calculate the corresponding anchor box value, enhance the feature learning ability of the network to bauxite, automatically learn the importance of different channel features, and improve the accuracy of bauxite target detection. In the experiment, 2189 bauxite photos were taken and screened as the target detection datasets, and the targets were divided into four categories: No. 55, No. 65, No. 70, and Nos. 72–73. By selecting the category volume balanced datasets, the optimal YOLOv4 network model was obtained after training 7000 times, so that the average accuracy of bauxite sorting reached 99%, and the reasoning speed was better than 0.05 s. Realizing the high-speed and high-precision sorting of bauxite greatly improves the mining efficiency and accuracy of the bauxite industry. At the same time, the model provides key technical support for the practical application of the same type of engineering.

**Keywords:** bauxite; K-means; SE; YOLOv4

## 1. Introduction

Bauxite, as the main raw material of alumina and metal aluminum, plays an irreplaceable role in the manufacturing fields of spacecrafts [1,2], automobiles [3,4], and so on. At the same time, due to the thermostability and wear resistance of bauxite, it has a wide range of application value in the fields of refractory [5,6], polishing powder [7], advanced grinding wheel, and so on.

At present, ore separation mainly depends on manual beneficiation and machine learning methods. In the traditional beneficiation process, ore separation mainly depends on the experience of professionals. Now, we use machine learning for ore separation. The intervention of professionals in ore separation is reduced, which not only improves the beneficiation capacity, but also reduces the process abnormality and equipment failure rate. The combination of convolution neural network and spectral technology [8–10], ore image segmentation [11], ABC-BP (Artificial Bee Colony-Back Propagation) neural network [12], and other improved methods [13–15] are used to realize the ore classification of image recognition and effectively solve the problem of manual separation in the process of ore production.

Traditional manual beneficiation has low separation efficiency and a serious waste of resources, which cannot meet the development needs of modern industry. The ore separation method based on machine learning improves the ore separation ability and solves some problems of traditional manual beneficiation, but the detection speed is not high and cannot achieve real-time detection in industry. For ore detection based on the

convolutional neural network, the corresponding region proposals on the bauxite photo are first generated and then a feature extraction and classification on the region proposals are carried out, which reduces the speed of ore detection. In order to improve the ore detection speed, the ore positioning and identification can be processed at the same time, so as to realize end-to-end optimization and significantly improve the detection speed.

Therefore, a new bauxite separation method is proposed in this paper. Aiming at the problem of insufficient detection accuracy of self-built bauxite datasets, an improved YOLOv4 network combining the SE (Squeeze-and-Excitation) attention module and the K-means clustering algorithm is proposed. The K-means clustering algorithm clusters bauxite in the datasets to determine the length–width ratio of bauxite. By adding the SE attention module to the YOLOv4 network, we can enhance the ability of the network to learn the characteristics of bauxite, automatically learn the importance of different channel characteristics, and improve the accuracy of bauxite target detection. It has potential application value in the fields of mining intelligence and protection of precious resources and provides a theoretical reference for further practical application.

## 2. Design of Bauxite Separation Model

In this section, we will show how to build a YOLOv4 network, leading to the K-means clustering algorithm and the SE attention module to establish an improved YOLOv4 network.

### 2.1. YOLOv4 Target Detection Algorithm

YOLOv4 [16] introduces the path aggregation network (PANet), spatial pyramid pooling (SPP), Mish activation function, and other technologies to improve the detection accuracy of targets. The backbone part adopts the CSPDarknet53 network that integrates the CSPNet (Cross Stage Partial Network) [17] and Darknet53 and can reduce the amount of calculation and maintain or even enhance the learning ability of the convolutional neural network. The CSPNet solves the gradient information repetition problem of network optimization in other large-scale convolutional neural network frameworks and integrates the gradient changes into the feature map from beginning to end, thus reducing the parameter amount and FLOPS value of the model, which not only ensures the inference speed and accuracy but also a reduced model size. The Neck part uses SPP [18] as an additional module to solve how the feature maps of different sizes enter the fully connected layer, which can greatly improve the receptive field of the network and separate the most significant upper and lower features. Using PANet [19] as the feature fusion module, a topdown and bottomup bidirectional fusion backbone network is proposed, and a "shortcut" is added between the bottom layer and the top layer to shorten the gap between layers. The path can repeatedly extract the features of the effective feature layer. The Head part is the head structure of YOLOv3, which extracts features from the feature layer for prediction. The network structure diagram of YOLOv4 is shown in Figure 1.

In Figure 1, the bauxite image is input into the backbone network to complete feature extraction, and then the fusion of feature maps of different scales is completed through SPP and PANet. Finally, the feature maps of three scales are output to predict the boundary box, class, and confidence.

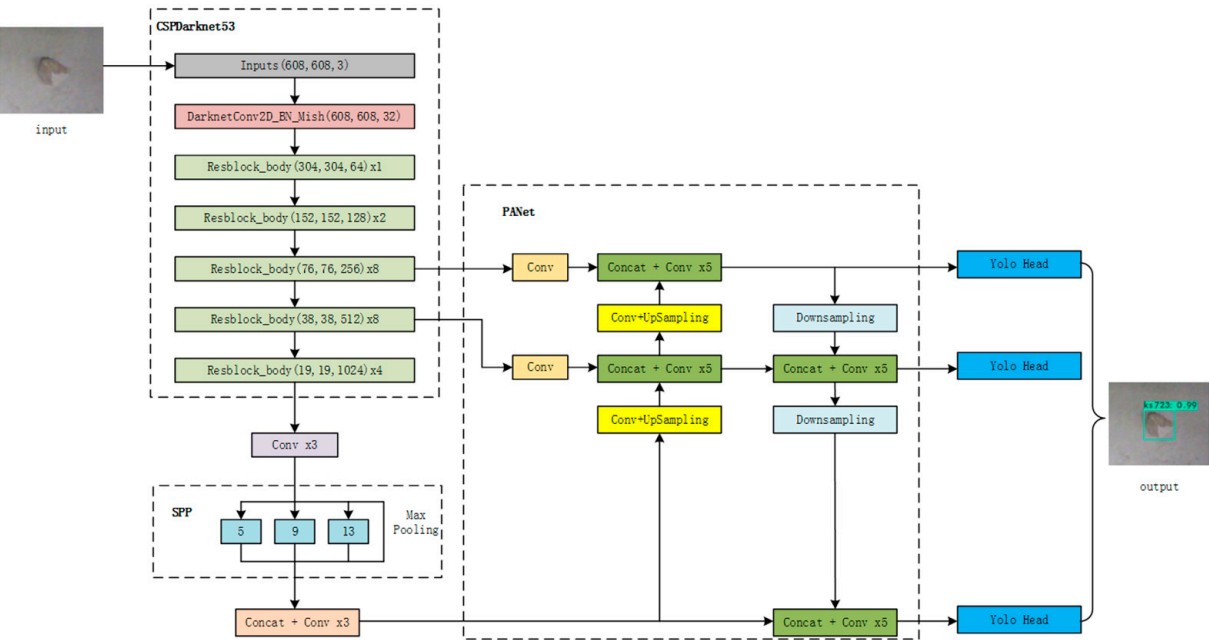

**Figure 1.** YOLO v4 network structure.

## 2.2. Improvement of YOLOv4 Algorithm

In the YOLOv4 network, nine anchor boxes can be preset to determine the length–width ratio of the detection target, and nine anchor boxes can be generated in each grid for detection to predict the bounding box of the target. In the detection of bauxite, the ore forms are different, and the anchor box default by YOLOv4 is not suitable for the detection of bauxite. Therefore, this paper uses the K-means clustering algorithm to cluster the bauxite size in the datasets and calculate the corresponding anchor box value.

The SE attention mechanism module [20], which screens out the attention for channels by learning the correlation among channels, can be easily embedded into the network model and only adds a small amount of model overhead and complexity. In the bauxite detection task, with the deepening of the training network, the bauxite characteristics gradually weaken, which can easily cause missed detection. However, embedding the SE attention module in the YOLOv4 network can enhance the learning ability of the network, automatically learn the importance of different channel characteristics, and improve the accuracy of bauxite sorting. As shown in Figure 2, the SE module is embedded in the Inception module to form a new SE–Inception module. In this study, the SE attention module is embedded into the Resunit module to form a new SE–Resunit module, and the SE module is embedded behind the CSPn module to form a new SE–CSPn module. The specific structure is shown in Figure 3.

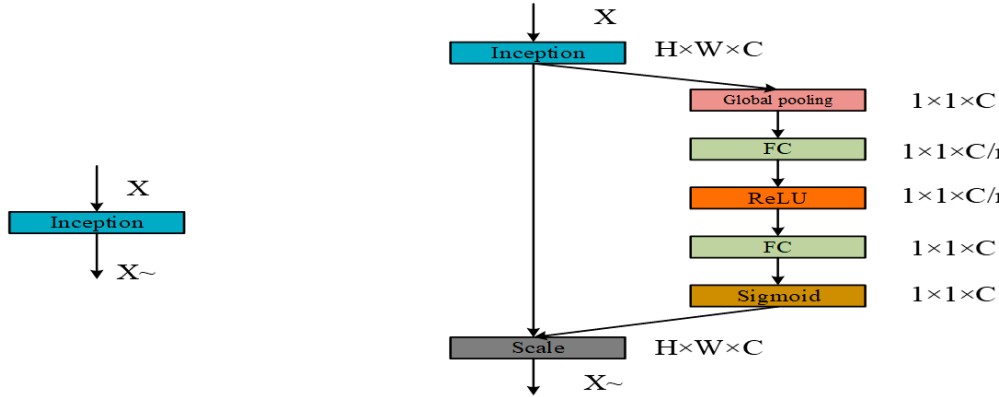

**Figure 2.** Original Inception module (**left**) and SE–Inception module (**right**).

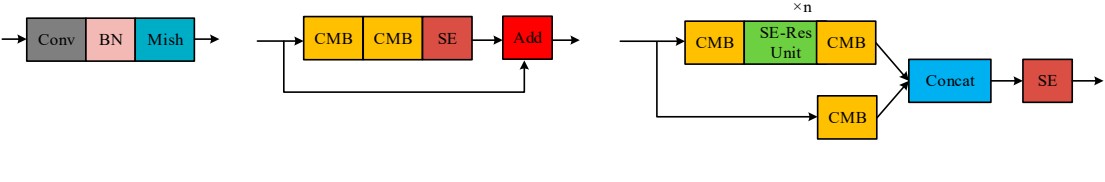

CMB module structure        SE-ResUnit module structure        SE-CSPn module structure

**Figure 3.** YOLOv4 network structure with SE module.

## 3. Loss Function of YOLOv4 Network

The loss function is used to measure the difference between the predicted value and the actual value. In this paper, the loss function is expressed as:

$$L_{CIOU} = 1 - IoU + \frac{\rho^2}{c^2} + \alpha v \tag{1}$$

The above loss function consists of three parts: the first part is the $L_{CIOU}$ loss function; the second part introduces the penalty term, and the third part is the width–height ratio of the frame where $IoU$ represents the degree of overlap between the prediction frame and the real frame in target detection; $\rho$ represents the Euclidean distance between the center point coordinates of the prediction frame and the real frame in target detection; $c$ represents the diagonal distance of the smallest box covering the prediction box and the real box; $\alpha$ is the weight function; $v$ is the consistency of measuring the width–height ratio.

### 3.1. $L_{IOU}$ Loss Function

From Equation (1), we know that the $L_{IOU}$ loss function is expressed as:

$$L_{IOU} = 1 - IoU \tag{2}$$

$$IoU = \frac{A \cap B}{A \cup B} \tag{3}$$

where $IoU$ represents the intersection and union ratio of the prediction frame and real frame in target detection. In Formula (3), $A$ represents the prediction frame, and $B$ represents the real frame. Obviously, the higher the value of $IoU$, the higher the degree of coincidence between the prediction frame and the real frame, the higher the prediction accuracy of the representative model, but the worse the performance of the representative model.

### 3.2. $L_{DIOU}$ Loss Function

The $L_{DIOU}$ loss function is composed of penalty term function based on the $L_{IOU}$ loss function. Its loss function is expressed as:

$$L_{DIOU} = 1 - IoU + \frac{\rho^2}{c^2} \tag{4}$$

The penalty term of the $L_{DIOU}$ loss function is based on the ratio of the distance between the center point and the diagonal, which avoids the large outsourcing frame when the distance between the two frames is far, resulting in the large value of the loss function, something that is difficult to optimize. Even when one box contains another box, the $c$ value remains unchanged, but the distance of the center point can be effectively measured.

### 3.3. Weight Function $\alpha$ and Width–Height Ratio $v$

When the center points of the two frames coincide, the values of $c$ and $\rho$ do not change; therefore, we introduced the weight function $\alpha$ and width–height ratio $v$, and the function expression is as follows:

$$\alpha = \frac{v}{(1 - IoU) + v} \tag{5}$$

$$v = \frac{4}{\pi^2}\left(\arctan\frac{w^{gt}}{h^{gt}} - \arctan\frac{w}{h}\right)^2 \qquad (6)$$

where $w^{gt}$ and $h^{gt}$ are the width and height of the real frame, and $w$ and $h$ are the width and height of the prediction frame. If the width and height of the real frame and the prediction frame are similar, that is, $v$ is 0, and this item is 0. The function of $\alpha*v$ is to control the width and height of the prediction frame to be as close as possible to the width and height of the real frame.

### 3.4. Mish Activation Function

The Mish activation function is used to add nonlinear factors, improve the model expression ability of the network, and solve the problems that cannot be solved by the linear network model. In this paper, we introduce the Mish activation function, whose expression is as follows:

$$Mish = x \times \tanh(\ln(1 + e^x)) \qquad (7)$$

The function image is shown in Figure 4.

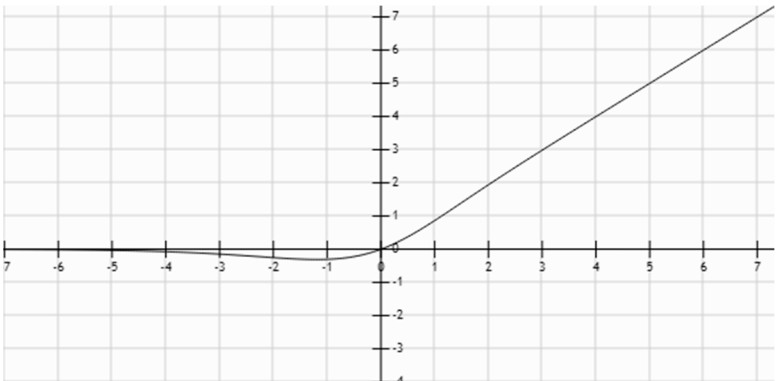

**Figure 4.** This Mish activation function image.

The *Mish* activation function has the following advantages: There is no upper boundary, which avoids the saturation caused by capping, and there is no gradient disappearance in the training process. Each point of the function is smoother, allowing better information to go deep into the neural network. When the value is negative, it allows a smaller negative gradient to flow in, ensuring that the information will not be interrupted, so as to obtain better accuracy and generalization ability.

## 4. Bauxite Separation Test Results and Discussion

This section introduces data preparation, establishment of control experiment, selection of optimal model, and experimental verification in the bauxite separation process.

### 4.1. Data Preparation

According to the research, we choose a 3D structured light depth camera Astra. According to Astra parameters, the data acquisition distance in this paper is determined to be 1.5 m to realize the data acquisition of four bags of sorted bauxite. According to the pictures collected in this paper, there is only one kind of bauxite in one picture. After all the pictures were taken, the total amount of data was finally sorted out, as shown in Table 1.

**Table 1.** Total data.

| Bauxite Category | Number of Pictures |
|---|---|
| No. 55(ks55) | 1000 |
| No. 65(ks65) | 400 |
| No. 70(ks70) | 479 |
| Nos. 72–73(ks723) | 310 |

The above datasets were labelled in LabelMe. After labeling, we integrated the datasets into different training sets and test sets according to the ratio of 8:2. When making the category volume of unbalanced datasets, a total of 2189 pictures were divided into 1751 training sets and 438 test sets. When making the category volume of balanced datasets, considering that the bauxite of Nos. 72–73 is the least, the final rounding was 250 as the training sets and 60 as the test sets. The other three types of bauxite also took out 310 pictures, including 1000 training sets and 240 test sets in the final category volume of balanced datasets. As shown in Table 2:

**Table 2.** Sorted datasets.

| Dataset Type | Number of Training Sets | Number of Test Sets |
|---|---|---|
| category volume unbalanced datasets | 1751 | 438 |
| category volume balanced datasets | 1000 | 240 |

*4.2. Experimental Process*

In this paper, in order to generate a better bauxite separation model, we used the improved YOLOv4 network and set up two groups of controlled experiments of the data categories balanced and unbalanced to find the optimal model. The accuracy P and the average AP of each category were used to judge the network performance. When the $IoU \geq 0.5$, the bauxite is detected correctly.

In the YOLOv4 network, both batch and subdivisions were set to 64, which means that batch pictures were loaded into the memory at one time during the training process, and then the forward propagation process was completed in subsets; the maximum number of iterations of Max_Batches was set to 8000; the number of detection categories was set to 4. The experimental equipment includes AMD Ryzen 5 3600X 6-Core Processor, GeForce RTX 2060 graphics card, and 16G memory.

Figure 5 shows the change of loss function of bauxite datasets in the process of the improved YOLOv4 network training. After 8000 training iterations, the loss function no longer shows a downward trend. In order to establish the best bauxite sorting model, we set 1000 ~ 8000 iterations for comparison, and then trained on the balanced and unbalanced bauxite category data, respectively. The training results are shown in Tables 3 and 4.

**Table 3.** Training results of improved YOLOv4 network for unbalanced category datasets.

| Number of Iterations | Bauxite Category | | | | mAP |
|---|---|---|---|---|---|
| | No. 55 | No. 65 | No. 70 | Nos. 72–73 | |
| 1000 | 97.59% | 47.98% | 77.31% | 48.04% | 67.82% |
| 2000 | 98.12% | 95.73% | 93.80% | 83.30% | 92.72% |
| 3000 | 98.14% | 99.19% | 99.18% | 96.01% | 98.13% |
| 4000 | 96.87% | 99.49% | 99.73% | 95.70% | 97.95% |
| 5000 | 97.51% | 98.76% | 99.80% | 93.20% | 97.32% |
| 6000 | 97.36% | 99.23% | 98.91% | 96.98% | 98.12% |
| 7000 | 97.34% | 98.68% | 99.87% | 95.13% | 97.75% |
| 8000 | 97.42% | 97.81% | 99.85% | 95.29% | 97.59% |

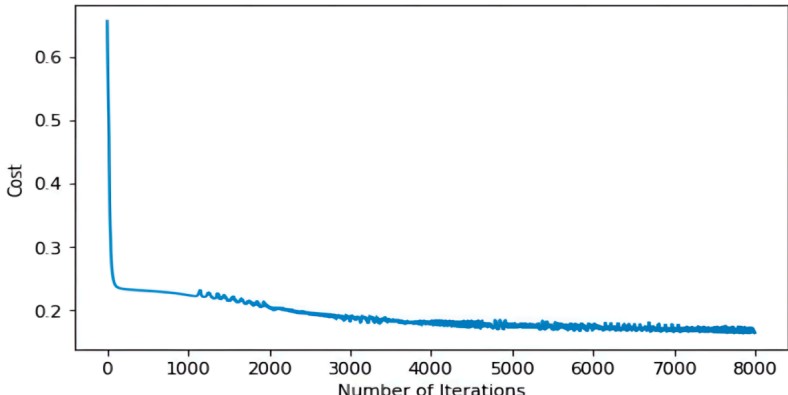

**Figure 5.** The abscissa represents the training iterations; the ordinate represents the total network loss.

**Table 4.** Training results of improved YOLOv4 network for balanced category datasets.

| Number of Iterations | Bauxite Category | | | | mAP |
|---|---|---|---|---|---|
| | No. 55 | No. 65 | No. 70 | Nos. 72–73 | |
| 1000 | 98.63% | 48.43% | 88.23% | 60.15% | 73.86% |
| 2000 | 97.94% | 98.18% | 98.55% | 93.65% | 97.08% |
| 3000 | 98.21% | 98.05% | 98.51% | 94.81% | 97.40% |
| 4000 | 98.20% | 98.18% | 98.55% | 96.55% | 97.87% |
| 5000 | 98.21% | 98.48% | 98.55% | 91.47% | 96.68% |
| 6000 | 98.21% | 99.81% | 98.53% | 93.80% | 97.59% |
| 7000 | 98.20% | 99.90% | 98.55% | 99.81% | 99.12% |
| 8000 | 98.21% | 98.48% | 98.55% | 93.09% | 97.08% |

In order to enhance the visibility of the data, we visualized the above table in Figure 6. The left figure shows the training results of the category volume unbalanced datasets in the improved YOLOv4 network, and the right figure shows the training results of the category volume balanced dataset in the improved YOLOv4 network. Through longitudinal comparison and observation, it is easy to conclude that given the limited sample data, when the datasets are made into a balanced category of data volume, the effect of the model with the same training times is better. When the model iterates 7000 times, the network training results of the four types of ores are the best, and the average detection accuracy is about 99%. The inference speed of the network model is between 0.03 s and 0.05 s, and the training effect is the best. Therefore, we selected the category volume balanced datasets to train the model for 7000 times under the improved YOLOv4. The confusion matrix obtained under this network model is shown in Figure 7.

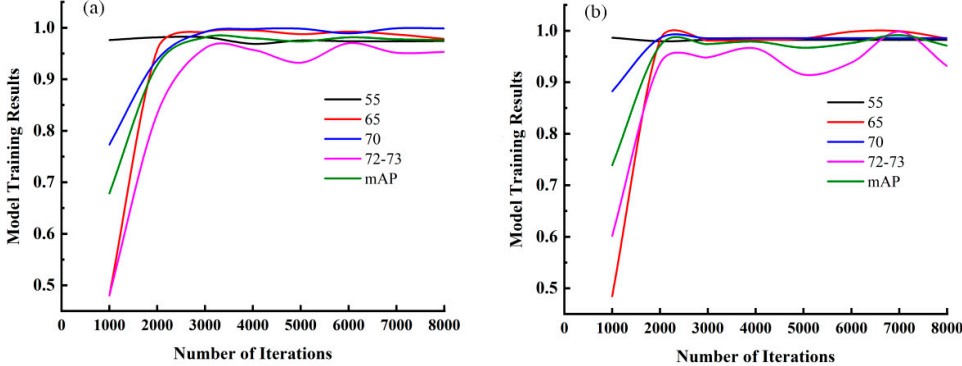

**Figure 6.** Abscissa represents the number of model iterations; ordinate represents the model training results, and different colors represent different types of bauxite. The left figure (**a**) represents the training results of the unbalanced datasets, and the right figure (**b**) represents the training results of the balanced datasets.

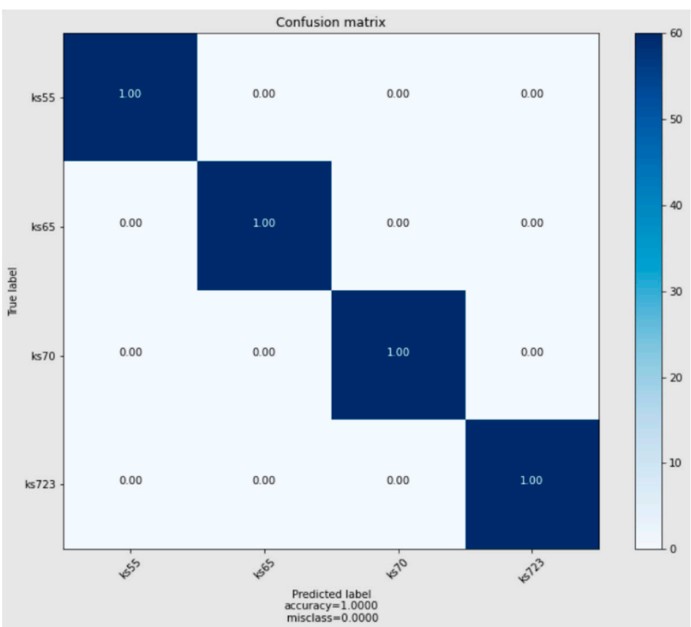

**Figure 7.** Confusion matrices obtained under the optimal model for the test sets of four ores. It can be seen from the figure that the detection sensitivity of the four types of ores is very high; the detection accuracy is almost 100%, and there is no misjudgment.

The same bauxite datasets are trained and verified on the original YOLOv4 network and the improved YOLOv4 network. Through analysis and calculation, it can be concluded that the average detection accuracy of the improved YOLOv4 network is increased by 10%; the false recognition rate is basically 0; the overall detection accuracy deviation is 1%, and the variance is almost 0.

*4.3. Experimental Verification*

After the above analysis, we obtained a bauxite sorting model based on the improved YOLOv4 network. In order to verify whether the model is reliable and whether the unknown pictures can accurately locate the bauxite and correctly identify it, we took some bauxite pictures for verification, and the results are shown in Figure 8. From left to right are the corresponding test results of No. 55, No. 65, No. 70, and Nos. 72–73, which can correctly identify the type of bauxite and correctly locate it. Through this test result, it is verified that the model of bauxite balanced datasets trained 7000 times under the improved YOLOv4 network has good bauxite sorting ability and can realize the real-time detection and classification of bauxite.

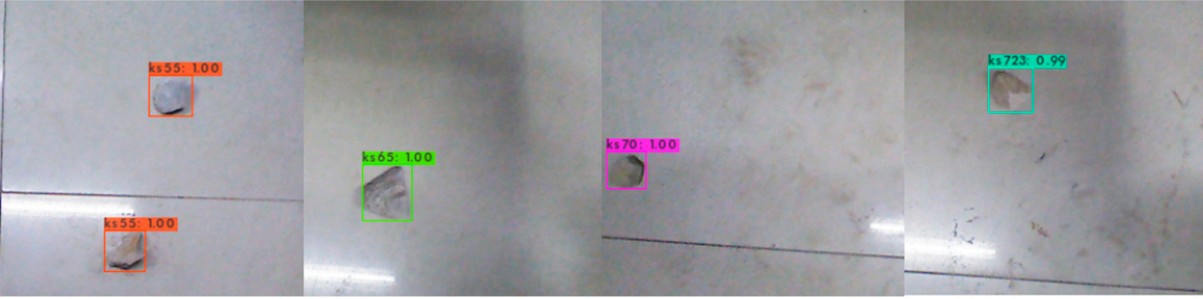

**Figure 8.** Corresponding test results of four bauxites.

**5. Conclusions**

Bauxite separation technology is of great significance in the fields of mining intelligence and resource protection. The problem of insufficient detection accuracy of self-built bauxite

datasets was studied. Based on the YOLOv4 network, the K-means clustering algorithm was used to cluster the bauxite size in the datasets to find the most suitable anchor box value. The SE attention module is embedded in the YOLOv4 network, so that the network can learn the correlation among channels, screen out the attention for channels, automatically learn the importance of different channel characteristics, and improve the detection accuracy of bauxite. The experimental results show that the detection accuracy of the improved YOLOv4 network for bauxite is as high as 99%, and the reasoning speed is between 0.03 s and 0.05 s, which can realize real-time detection. Compared with the original YOLOv4 network, the average detection accuracy of the improved YOLOv4 network is increased by 10%; the false recognition rate is basically 0; the overall detection accuracy deviation is 1%, and the variance is almost 0. Finally, the effectiveness of the improved algorithm is also verified. The bauxite detection method proposed in this paper effectively solves the problem of bauxite sorting and lays a solid foundation for the application of bauxite in various fields. However, the improved YOLOv4 model has not yet carried out the classification and detection of other ores. We will further study the universality of ore detection of this model. It provides a theoretical reference for further practical application.

**Author Contributions:** Conceptualization, P.Z. and B.Z.; methodology, Z.L.; software, J.L.; validation, Y.L., Z.L. and J.L.; formal analysis, P.Z.; investigation, P.Z.; resources, B.Z.; data curation, Z.L.; writing—original draft preparation, Z.L.; writing—review and editing, P.Z.; visualization, Y.L.; supervision, B.Z.; project administration, P.Z.; funding acquisition, B.Z. All authors have read and agreed to the published version of the manuscript.

**Funding:** This research was funded by the National Natural Science Foundation of China (NSFC) (62175219).

**Institutional Review Board Statement:** Not applicable.

**Informed Consent Statement:** Not applicable.

**Data Availability Statement:** Not applicable.

**Conflicts of Interest:** The authors declare no conflict of interest.

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
