# Peer review of "Machine Learning Sorting Method of Bauxite Based on SE-Enhanced Network"

_applsci, doi:10.3390/app12147178_

Round 1
Reviewer 1 Report
· This manuscript should be revised based on the remarks given below
n in line, 38 the original name of the abbreviations should be given once.
Reference should be made to the information given on lines 41 and 42.
In line 53, SE should be given once - Squeeze-and-Excitation
In the data preparation paragraph (section 4.1), the clustering process is not given clearly. If clustered by aspect ratio only, clusters can be presented visually on a single graph. Also, what is the reason for choosing 4 clusters? Was it decided by elbow, silhouette or some other method, or were 4 clusters selected directly? Are there any other characteristic differences between the groups?
In line 226, the figure number was given incorrectly, 2 was written instead of 6. It should also be specified as a balanced and unbalanced data set on the figure.
In line 242-242, sentence should be revised.
n general, Bauxite identification/classification has been made with machine learning and the purpose is beneficiation. However, the particle size used is not given. What are the beneficiation methods in the relevant sizes? Comparison with machine learning based sorting and these methods can be given. Emphasis was made on bauxite enrichment in the introduction, but it was not sufficiently mentioned in the conclusion.
Reviewer 2 Report
A very interesting issue and it was presented in a transparent manner.
Please consider extending section 2.1 to introduce the used methods and algorithms more specifically. Currently, section 2.1 may be difficult to understand for people who do not deal with deep learning on a daily basis.
When using abbreviations for the first time, please enter the full name, e.g. SE, ABC-BP neural network.
Please check the document, there seem to be some unnecessary spaces in it, e.g. section 4.1.
In section 4.2 please consider adding bauxite detection sensitivity. The ROC curve is also a good measure to judge the quality of the model.
In the text, there is a lack of technical information about the computers on which the algorithm was tested. Only the time difference (line 250) is stated in the conclusions section. There is no detailed comparison between balanced and unbalanced datasets. The article would also gain benefit from a broader comparison between the original method and its modification.
